# Naxitamab Combined with Granulocyte-Macrophage Colony-Stimulating Factor as Consolidation for High-Risk Neuroblastoma Patients in First Complete Remission under Compassionate Use—Updated Outcome Report

**DOI:** 10.3390/cancers15092535

**Published:** 2023-04-28

**Authors:** Jaume Mora, Alicia Castañeda, Maite Gorostegui, Amalia Varo, Sara Perez-Jaume, Margarida Simao, Juan Pablo Muñoz, Moira Garraus, Cristina Larrosa, Noelia Salvador, Cinzia Lavarino, Lucas Krauel, Salvador Mañe

**Affiliations:** Pediatric Cancer Center Barcelona, Hospital Sant Joan de Déu, 08950 Barcelona, Spain

**Keywords:** neuroblastoma, naxitamab, GM-CSF, high-risk, consolidation, anti-GD2 immunotherapy

## Abstract

**Simple Summary:**

Naxitamab (Danyelza®) is a newly FDA-approved humanized anti-GD2 antibody for the treatment of relapsed/refractory in the bone/bone-marrow-only compartment high-risk neuroblastoma. In our center, we had the unique opportunity to use Naxitamab in patients achieving complete remission with standard management including autologous stem cell transplantation or not. In 2021, we reported the first cohort ever of such patients (*n* = 55) and showed very encouraging survival results at 3 years. Hereby, we present an updated report on the outcome of a larger cohort (*n* = 82) of patients followed for significantly longer (a median follow-up of 37 months). The results demonstrate reassuring survival rates (5-year event-free survival of 57.9% and overall survival of 78.6%) for high-risk neuroblastoma patients achieving end-of-induction complete remission. This study adds to increasing evidence that high-dose chemotherapy with an autologous stem cell transplant may not be required to achieve long-term survival in, at least, this subgroup of neuroblastoma patients.

**Abstract:**

Naxitamab is an anti-GD2 antibody approved for the treatment of relapsed/refractory HR-NB. We report the survival, safety, and relapse pattern of a unique set of HR-NB patients consolidated with naxitamab after having achieved first CR. Eighty-two patients were treated with 5 cycles of GM-CSF for 5 days at 250 μg/m^2^/day (−4 to 0), followed by GM-CSF for 5 days at 500 μg/m^2^/day (1–5) and naxitamab at 3 mg/kg/day (1, 3, 5), on an outpatient basis. All patients but one were older than 18 months at diagnosis and had stage M; 21 (25.6%) pts had MYCN-amplified (A) NB; and 12 (14.6%) detectable MRD in the BM. Eleven (13.4%) pts had received high-dose chemotherapy and ASCT and 26 (31.7%) radiotherapy before immunotherapy. With a median follow-up of 37.4 months, 31 (37.8%) pts have relapsed. The pattern of relapse was predominantly (77.4%) an isolated organ. Five-year EFS and OS were 57.9% (71.4% for MYCN A) 95% CI = (47.2, 70.9%); and 78.6% (81% for MYCN A) 95% CI = (68.7%, 89.8%), respectively. EFS showed significant differences for patients having received ASCT (*p* = 0.037) and pre-immunotherapy MRD (*p* = 0.0011). Cox models showed only MRD as a predictor of EFS. In conclusion, consolidation with naxitamab resulted in reassuring survival rates for HR-NB patients after end-induction CR.

## 1. Introduction

The curing of high-risk (HR) neuroblastoma (NB) patients remains challenging. Standard treatments comprise a backbone of induction chemotherapy and surgery, followed by consolidation with high-dose chemotherapy and an autologous stem cell transplant (ASCT), and post-consolidation therapy with radiotherapy and anti-GD2-based immunotherapy plus cis-retinoic acid (Cis-RA).

Initial studies suggested myeloablative therapy in conjunction with autologous bone marrow (BM) transplant might provide an improved outcome [1,2,3,4,5,6,7,8,9,10,11,12,13,14,15]. Dose-intensive therapies appeared to be the common denominator of all these studies. Cheung and Heller in 1991 [16] performed a large retrospective meta-analysis and concluded that the intensification of etoposide and platinum agents had the greatest influence on response, event-free survival (EFS), and overall survival (OS) in neuroblastoma. Doxorubicin and cyclophosphamide had less of an effect and vincristine intensity had no effect. Twenty-one weeks of treatment duration seemed to be sufficient, beyond which there was no clear gain. The strongest correlation of dose intensity was EFS, and there was no sign that EFS or OS curves would merge with increasing dose intensification. These mathematical findings raised the question as to whether a cure was possible with only cytotoxic (chemo- and radio-) therapy. The role of dose intensification in NB was tested in several subsequent studies (reviewed in [15]) and, importantly, in three randomized trials [17,18,19]. All three randomized trials concluded, as predicted, modest improvements in EFS but not in OS. The same findings were obtained when investigating increased dose intensity during induction; a single transplant versus none; or a tandem transplant versus a single transplant [20]. At the same time, a confirmatory meta-analysis from 2015 [21] found no differences in OS.

It appeared that a cure was hard to reach for most HR-NB cases with therapies based solely on chemotherapy, surgery, and radiotherapy. Fortunately, chemo-resistant NB was found to be highly responsive to anti-GD2-based immunotherapy [22] with or without chemotherapy [23]. Notably, two anti-GD2 monoclonal antibodies (mAbs)—dinutuximab and naxitamab—have been approved by the US FDA in the last 7 years. Similar to ASCT, anti-GD2 immunotherapy aims to eradicate chemotherapy-resistant diseases. Clinical trials using anti-GD2 immunotherapy demonstrated a significant improvement in EFS and OS [24,25]. However, both strategies aiming to eradicate minimal residual disease (MRD), i.e., ASCT and anti-GD2 immunotherapy, have not been tested separately from each other. Indeed, the best outcomes reported to date for children with HR-NB from cooperative groups is the recent update from COG of stage 4 patients treated with ANBL0032 describing 5-year EFS and OS values of 57 and 70.9%, respectively [26]. These results were from stage 4 patients older than 12 months at diagnosis treated with dinutuximab-based immunotherapy after ASCT having shown a pre-ASCT (end-induction) objective response: Complete response (CR), very good partial response (VGPR), or partial response (PR) by 1993 International Neuroblastoma Response Criteria (INRC) [9] for the primary site, soft tissue metastases, and bone metastases. Furthermore, patients could have <10% tumor from a bone marrow aspirate/biopsy or newly detected marrow disease if the extent of tumor involvement was <10%. Reported outcomes (5-y EFS and OS) according to the pre-ASCT response (excluding bone marrow) for CR patients (*n* = 352; 29.8%) were 64.2 and 72.7%; for VGPR patients (*n* = 418; 35.3%), they were 59.7 and 68.2%; and for PR patients (*n* = 413; 34.9%), they were 55.4 and 70.5%, respectively. This is the current benchmark for the best subgroup of HR-NB cases, those achieving CR at end-induction. In contrast, the Memorial Sloan–Kettering Cancer Center (MSKCC) team in 2016 reported the first cohort of HR-NB patients consolidated with anti-GD2 immunotherapy without ASCT [27]. The 5-year rates for the CR/VGPR patients at end-induction managed without ASCT were EFS 51% and OS 75%, which were statistically not significantly different compared to those having received ASCT. This was the first reported clinical observation that suggested that anti-GD2 immunotherapy could be sufficient to consolidate patients achieving CR/VGPR at the end of induction.

In 2021, we reported a cohort of 55 patients with HR-NB in first CR at end-induction consolidated with naxitamab and GM-CSF [28]. Our analysis showed a 3-year EFS of 74.3% and OS of 91.6%. Prognostic markers relevant for HR-NB patients such as BM MRD prior to immunotherapy [29,30,31] or MYCN status [32] were not significant. High-dose chemotherapy and ASCT or radiation therapy prior to immunotherapy did not show statistical significance for survival. In this study, we expanded the cohort to 82 HR-NB patients and updated the survival, safety, and pattern of relapse of HR-NB patients achieving first CR at end-induction consolidated with naxitamab-based immunotherapy. Models were built to test for prognostic variables that might affect outcomes.

## 2. Materials and Methods

We report on 82 consecutive patients in first CR (defined according to the INRC criteria by Park et al. 2017 [33]) treated at Hospital Sant Joan de Déu (HSJD) with naxitamab and granulocyte-macrophage colony-stimulating factor (GM-CSF) under compassionate use from June 2017 until April 2022. This study included patients with HR-NB (stage M at age > 18 months or *MYCN*-amplified at any age or stage) who achieved first CR after receiving induction chemotherapy regimens including alkylators and platinum-containing compounds (at least 5 cycles) and surgery. The list of induction regimens used at local institutions includes mN7, SIOPEN HR-NB01, GPOH NB2004, Chinese BCC trials, GALOP trials, COG 3973, and CCG regimens. Consolidation included radiotherapy but ASCT was not required.

Patients received naxitamab-based immunotherapy under compassionate use if CR was confirmed in the extent of the disease work-up, the diagnosis was reviewed and confirmed, and major organ toxicity from prior therapies was grade ≤2 by Common Terminology Criteria for Adverse Events (CTCAE) Version 4.0. Informed written consent for treatments and tests was obtained according to HSJD institutional review board rules.

Radiation delivered to the primary and bulky sites at diagnosis was administered as previously described [28] either before immunotherapy (for *MYCN*-amplified cases mostly), after induction (chemo and surgery and ASCT if the patient received it in their local institution), or after immunotherapy for *MYCN* non-amplified cases.

Naxitamab-based immunotherapy cycles comprised priming doses of subcutaneous (SC) GM-CSF for 5 days at 250 μg/m^2^/day (days −4 to 0), followed by SC GM-CSF for 5 days at 500 μg/m^2^/day (days 1–5) and naxitamab infused intravenously (IV) over 30–120 min, at 3 mg/kg/day on days 1, 3, and 5, for a total dose of 9 mg/kg (~270 mg/m^2^) per cycle. GM-CSF was not given if the neutrophil count was >20,000/μL or total white blood cells > 50,000/μL. Immunotherapy treatment cycles were repeated every 4 weeks (±1 week) for a total of 5 cycles. Naxitamab treatment was administered in an outpatient setting (day hospital) in all cases.

As previously reported, all patients received daily oral docosahexaenoic acid triglyceride (DHA-TG) at 0.25 g/kg, half administered in a single oral intake and the rest in the other two administrations during the day, matched with meals, instead of cis-RA. DHA-TG is labeled as “food for special medical purposes” and not a medicinal product. The intervention is considered dietary supplementation. The rationale for using DHA in neuroblastoma patients is supported by studies showing how, compared to normal nervous tissues, neuroblastoma is profoundly deficient in DHA-TG, whereas the level of Omega-6 fatty acid arachidonic acid is increased [34]. These factors suggest that an imbalance between Omega-3 and Omega-6 fatty acids may serve as an adaptation mechanism in neuroblastoma. DHA-TG competes with the Omega-6 fatty acid arachidonic acid for conversion by cyclooxygenases and lipooxygenases to a wide range of lipid mediators, causing a plethora of physiological effects on the organism. In fact, it is demonstrated that DHA-TG induces apoptosis of neuroblastoma cells by mechanisms involving the intracellular accumulation of cytotoxic DHA-derived metabolites [35,36], and treatment with DHA in murine xenografts with human neuroblastoma cells resulted in stable disease or a partial response, depending on the DHA concentration [36].

Adverse events were prospectively collected and reported in parallel to authorities and Ymabs, according to established agreements between HSJD and Ymabs Therapeutics. CTCAE version 4.0 was used for reporting.

### 2.1. Disease Evaluations and Treatment Monitoring

As previously reported [28], disease status was thoroughly assessed by BM aspirates obtained from bilateral posterior and bilateral anterior iliac crests, 123I-MIBG SPECT scans, and whole-body and craniospinal MRI. FDG-PET was used for MIBG non-avid cases at diagnosis. Disease response was defined according to the revised INRC [33]. BM aspirates and 123I-MIBG/FDG-PET scans were performed after cycles 2 and 5 to evaluate the response. Treatment continued if CR was reconfirmed after cycle 2, for a total of 5 planned cycles.

A quantitative reverse transcription-polymerase chain reaction was used to assess minimal residual disease (MRD) status, as described previously [37]. During the follow-up, the disease status was assessed every 3 months for 2 years by BM aspirates (x4) and MRD in addition to 123I-MIBG/FDG-PET scans. Once a year, a craniospinal MRI was added.

The human antihuman antibody (HAHA) was quantified by an ELISA assay validated for its sensitivity, specificity, and interference as previously reported [22]. A cut point for HAHA positivity was set at 1300 U/mL in the original studies [22].

### 2.2. Statistical Analysis

As previously reported [28], continuous variables were described using the median, minimum, maximum, and categorical variables by absolute frequencies and percentages. Event-Free-Survival (EFS) was defined as the time from immunotherapy onset to progressive disease (PD), relapse, secondary malignancy, or death, whichever occurred first, and was censored at the last follow-up in the absence of events. Overall Survival (OS) was defined as the time from immunotherapy to death and was censored at the last follow-up if no death occurred. The Kaplan–Meier method [38] was used to estimate the EFS and OS curves. The prognostic impact of clinical and biological features on survival (either EFS or OS) was tested by the log-rank test [39]. Optimal cut-offs for age were estimated using the Contal–O’Quigley method [40] with OS as the outcome. Cox models [41] were used to derive hazard ratios (HR) for clinical and biological features on EFS and OS.

## 3. Results

A total of 82 consecutive HR-NB patients having confirmed first CR were treated with naxitamab and GM-CSF at HSJD. Table 1 describes the clinical and biological characteristics of all patients treated. All but one patient were >18 months of age and had stage M at diagnosis. The median age at diagnosis for the entire cohort is 3.0 years (range, 0.6–13) and 3.9 years at immunotherapy initiation. Twenty-one (25.6%) of the patients had *MYCN*-amplified (A) tumors. Eleven (13.4%) patients had ASCT after having achieved CR at their end-induction evaluation and before receiving immunotherapy. Twelve (14.6%) patients had positive BM MRD before immunotherapy. None of the patients had received anti-GD2 immunotherapy previously. The median time from diagnosis to immunotherapy initiation in this cohort is 8.7 months (range = 5.4, 21.5).

Sixty-seven (81.7%) of the eighty-two patients completed the total number of planned cycles. Among the 15 patients who did not complete the planned treatment, 4 experienced grade 4 toxicities: Two episodes of apnea; one opioid-related chest rigidity syndrome; and one stroke after completing the first cycle, which was extensively investigated and found to be unrelated to naxitamab. Eleven (13.4%) patients relapsed on treatment.

Patients who maintained complete remission at the end of consolidation (immunotherapy and radiotherapy) did not receive any further treatment. Figure 1 shows the Kaplan–Meier curves for the whole population from the time of immunotherapy, with 5-year EFS and OS values of 57.9% (71.4% for MYCN A) 95% CI = (47.2, 70.9%) and 78.6% (81.0% for MYCN A) 95% CI = (68.7%, 89.8%), respectively.

Table 2 lists all the 3-year and 5-year EFS and OS values and comparisons performed using the log-rank test. EFS showed significant differences between the cohort of patients having received prior ASCT (log-rank test, *p* = 0.037) and detectable MRD before immunotherapy (log-rank test, *p* = 0.0011), whereas OS was not significantly different in any of the groups.

Figure 2 shows the Kaplan–Meier curves showing the significant differences for EFS (and not for OS) between patients with positive or negative MRD, and Figure 3 shows patients having received ASCT prior to immunotherapy or not.

With a median follow-up for living patients of 37.4 (range, 9.1 to 66.6) months, 31 (37.8%) patients had relapsed during or after treatment. The pattern of relapse observed is predominantly (24/31; 77.4%) an isolated organ, primarily bone (14/31) and soft tissue (8/31). Among the 14 bone relapses, 7 (50%) had 3 or fewer bones affected. Three (3.7%) patients relapsed in the central nervous system (CNS), all from the *MYCN* A group.

The survival analysis using univariate Cox models including the same variables analyzed in the previous cohort, i.e., *MYCN* status, the number of chemotherapy cycles during induction, ASCT and/or radiotherapy before immunotherapy, MRD status at enrollment, and age at diagnosis or at immunotherapy initiation, is summarized in Table 3. None of the variables analyzed showed significance for predicting OS. Patients with detectable MRD prior to immunotherapy had a significant inferior EFS with a notable HR of 3.29 (*p* = 0.0020) compared to negative MRD patients. Noticeably, prior ASCT did not show statistical significance in predicting EFS in the Cox analysis. Table 3 lists the hazard ratios for all the variables analyzed. Multivariate analyses were not conducted since only one variable (MRD) was consistently significant in the univariate analyses for EFS and none were statistically significant in the univariate analyses for OS.

The human antihuman antibody was measured after every cycle. Human antihuman antibody positivity according to large previous experience (serum levels > 1300 U/mL) from MSKCC developed at some point during treatment in 6 of the 52 tested patients (11.5%), a similar range reported in the Phase 1 study of hu3F8 (16% after 2 cycles) [22].

## 4. Discussion

We report on a large cohort of patients with HR-NB in first complete remission after standard induction regimens (managed in local institutions) consolidated with the anti-GD2 mAb naxitamab and GM-CSF at HSJD. The analysis shows an encouraging survival rate with a 5-year EFS value of 57.9% and OS of 78.6%. Our previous report in 2021 did not show the prognostic significance of BM MRD or ASCT prior to immunotherapy [28]. However, in this updated analysis with an increased number of cases and a longer follow-up, MRD prior to immunotherapy was statistically significant in the two statistical tests performed to predict EFS. Patients having received ASCT prior to naxitamab immunotherapy showed a significantly improved 5-year EFS compared to patients who did not receive high-dose chemotherapy and ASCT in the log-rank test (but not in the Cox analysis), reproducing what has been reported in the literature in all studies where ASCT has been evaluated [17,18,19,20]. Patients with detectable MRD in the BM prior to immunotherapy had a significantly worse 5-year EFS compared to those with undetectable MRD.

Achieving CR after induction chemotherapy and surgery is the most consistent and reproducible prognostic variable in all reported neuroblastoma studies to date [42,43,44,45]. Therefore, the role of induction chemotherapy and surgery in the entire management of HR-NB is demonstrated [45,46]. The prognostic significance of MRD and ASCT prior to immunotherapy suggests that the quality of complete remission status is relevant, and thus the intensity of the induction regimens is critical. In this study, patients received different induction regimens at their local institutions before being referred to HSJD for immunotherapy consolidation. When reviewed, all MRD-positive patients before immunotherapy were managed in centers following reported regimens from the 1990s whereby dose intensity is lower than the current induction regimens used in Western Europe or North America. Since dose intensity has been correlated with outcomes (EFS) in neuroblastoma, it seems imperative to promote the most intensified induction regimens, whenever supportive therapy is available, to be used from the very beginning in the course of patient management to achieve the best outcomes. Furthermore, given the significance of MRD to predict relapse in our series, systematic determination of BM MRD at the end of induction may provide a relevant biomarker to improve the chances of decreasing relapses by introducing tailored strategies to clear MRD during induction. Among these, more cycles of intensified chemotherapy (if less than 7), increasing the number of anti-GD2 mAb cycles (the majority of relapses occurred after completing five cycles), or using different chemo-immunotherapy strategies are now being investigated as strategies.

The tailored management of patients with *MYCN*-amplified tumors would also be desired. Achieving first CR is the only real chance to cure *MYCN*-amplified HR-NB patients as they are virtually incurable when becoming refractory or after relapse [47,48]. In contrast, the MSKCC experience showed that *MYCN*-amplified tumors in first CR have better survival than *MYCN* non-amplified cases [32]. In our cohort, patients with *MYCN*-amplified tumors had a better 5-year EFS (71.4%) and OS (81%) compared to *MYCN* non-amplified cases, although the difference was not statistically significant given the low numbers. It is notable that among *MYCN*-amplified cases of the six relapses, three (50%) occurred in the central nervous system. These results suggest that patients with *MYCN*-amplified tumors may need specific *CNS* surveillance programs (during surveillance post-treatment, we routinely add an annual craniospinal MRI to standard MIBG), and, importantly, strategies to prevent *CNS* penetration should be explored.

The most recent update from COG for stage 4 patients older than 18 months of age at diagnosis having achieved CR before ASCT showed 5-year EFS of 64.2 and OS of 72.7% [26]. Our limited single-institution study showed 5-year EFS of 51.9% and OS of 76.9% survival rates for patients treated with naxitamab and GM-CSF without the use of high-dose chemotherapy and ASCT or cis-RA. These results are almost identical to those reported previously by the MSKCC team (EFS 51% and OS 75% [27]) and continue to question the need for ASCT and cis-RA in the overall management of HR-NB patients having achieved CR at the end of induction in the current era of anti-GD2 immunotherapy. If these results could be reproduced in larger cohorts and prospective clinical trials, they may prompt a new era of protocol standards with reduced costs, shorter durations of treatment, and most importantly, fewer long-term toxicities for those patients now having higher chances of survival.

The toxicity encountered in this large cohort of patients has provided a great deal of knowledge on the overall management of naxitamab [49]. Four patients experienced grade 4 toxicities that prevented planned therapy as previously reported [28], including two apnea, one chest rigidity syndrome (a secondary effect of opioids), and one patient with stroke, the causality of which related to naxitamab was not proven. Better protocols to manage acute toxicities and more adaptive infusion regimens have been developed to increase patient safety [49].

As per standard consolidation management, all patients in this study received radiotherapy of the primary tumor site, bulky sites (>5 cm) at diagnosis, or slow-responding sites of disease during induction. We analyzed if radiotherapy administered before (primarily *MYCN*-amplified tumors in our series) or after immunotherapy had an impact on survival or modified the relapse pattern. As previously reported [28], our confirmatory results do not demonstrate differences between patients receiving radiotherapy before or after immunotherapy.

A recurrent observation across anti-GD2 immunotherapy studies [24,25,26,28,31,50] or the GD2/GD3 vaccine [51] is the discrepancy in the impact on overall survival versus EFS (mostly relapse). This is quite striking in oncology since relapse does not necessarily imply a death sentence. The reason patients do not die despite disease recurrence is unclear yet. One of the reasons suggested previously is that the pattern of relapse after anti-GD2 immunotherapy tends to be more isolated and thus more amenable to rescue treatments, thus explaining, in part, the differential effect on survival [30]. Indeed, in this study, a limited relapsed pattern is clearly shown to be potentially easier to manage with modern re-induction treatments including re-challenges with the same or other anti-GD2 mAbs. Another potential reason to explain such a difference in outcome is the vaccination effect of an anti-idiotype network described previously for the murine mAb m3F8 [52]. The generation of an anti-idiotype network has not been demonstrated for all anti-GD2 antibodies and therefore this hypothesis needs further investigation. Similarly, whether the prolongation of treatment with anti-GD2 antibodies might increase the vaccination effect and EFS is also uncertain. However, early experiences with vaccination strategies might suggest that prolongation of the immune stimuli to keep anti-GD2 titers high could prevent relapses in patients who, otherwise, have a high relapse rate, such as those who have had a previous relapse [51].

## 5. Conclusions

Consolidation with naxitamab and GM-CSF immunotherapy has resulted in encouraging survival results for high-risk neuroblastoma patients in first complete remission. This study adds to the increasing evidence that high-dose chemotherapy with an autologous stem-cell transplant may not be required to achieve long-term survival in, at least, this subgroup of patients.

## Figures and Tables

**Figure 1 cancers-15-02535-f001:**
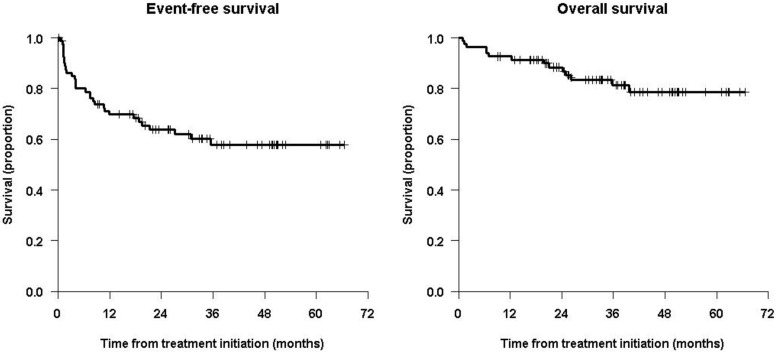
Kaplan–Meier survival curves for the whole study population from time of immunotherapy.

**Figure 2 cancers-15-02535-f002:**
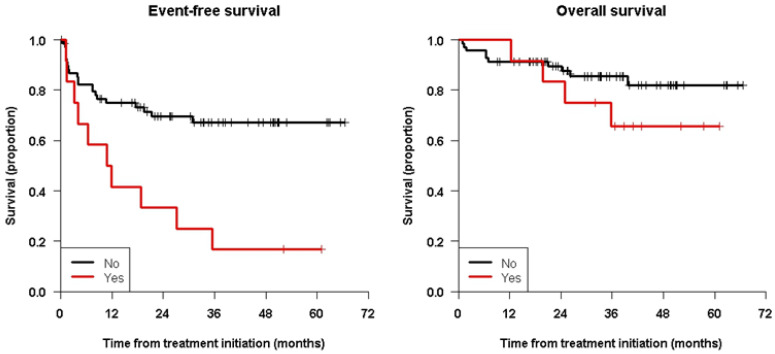
Kaplan–Meier survival curves for patients with positive or negative MRD before immunotherapy showing the significant differences for EFS but not for OS.

**Figure 3 cancers-15-02535-f003:**
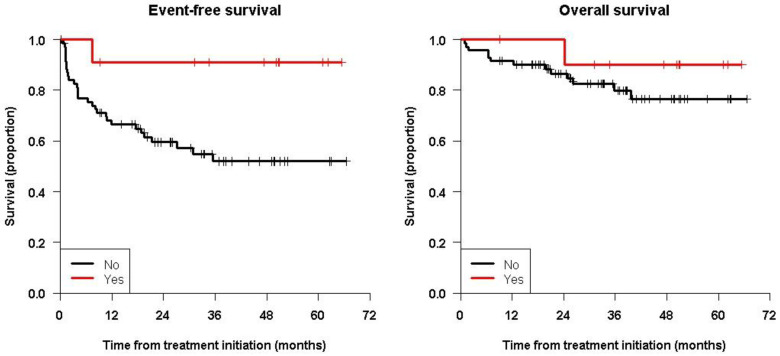
Kaplan–Meier survival curves for patients having or not received ASCT before immunotherapy showing the significant differences for EFS but not for OS.

**Table 1 cancers-15-02535-t001:** Summary descriptives of clinical and biological characteristics of all patients treated.

	*n* = 82	*n*
MYCN:		82
A	21 (25.6%)	
NA	61 (74.4%)	
Chemotherapy cycles:		82
5	13 (15.9%)	
>5	69 (84.1%)	
ASCT:		82
No	71 (86.6%)	
Yes	11 (13.4%)	
RDT:		82
No	56 (68.3%)	
Yes	26 (31.7%)	
MRD:		82
No	70 (85.4%)	
Yes	12 (14.6%)	
Age at diagnosis (y)	3.0 [0.6; 13.0]	82
Age at treatment initiation (y)	3.9 [1.4; 13.8]	82
Stage:		82
3	1 (1.2%)	
4	81 (98.8%)	
Induction Regimen:		82
mN7	16 (19.5%)	
GPOH NB2004	6 (7.3%)	
SIOPEN HR-NB-01	10 (12.2%)	
COG 3973	5 (6.1%)	
Chinese protocols	40 (48.8%)	
Russian HR-NB protocols	1 (1.2%)	
GALOP protocol	2 (2.4%)	
CCG	2 (2.4%)	

Legend: MYCN status: A = amplified; NA = non-amplified. ASCT = autologous stem-cell transplantation. MRD = minimal residual disease (Bone marrow). Age at DX = age at diagnosis; Age at Treat. = age at immunotherapy initiation; all in y = years.

**Table 2 cancers-15-02535-t002:** 3-year and 5-year EFS and OS with comparisons using the log-rank test.

	EFS			OS		
	3-Year Survival [95% CI]	5-Year Survival [95% CI]	*p*-Value	3-Year Survival [95% CI]	5-Year Survival [95% CI]	*p*-Value
All patients	57.9 [47.2; 70.9]	57.9 [47.2; 70.9]	-	81.3 [72.4; 91.3]	78.6 [68.7; 89.8]	-
MYCN: A	71.4 [54.5; 93.6]	71.4 [54.5; 93.6]	0.38	81.0 [65.8; 99.6]	81.0 [65.8; 99.6]	0.60
NA	53.3 [41.0; 69.2]	53.3 [41.0; 69.2]		81.7 [71.3; 93.5]	78.4 [67.0; 91.7]	
Chemotherapy cycles: 5	59.8 [37.8; 94.7]	59.8 [37.8; 94.7]	0.93	76.9 [57.1; 100.0]	76.9 [57.1; 100.0]	0.49
>5	57.6 [46.0; 72.1]	57.6 [46.0; 72.1]		82.0 [72.3; 93.1]	78.8 [67.8; 91.5]	
ASCT: No	51.9 [40.3; 66.9]	51.9 [40.3; 66.9]	0.037	79.9 [70.0; 91.2]	76.4 [65.2; 89.5]	0.36
Yes	90.9 [75.4; 100.0]	90.9 [75.4; 100.0]		90.0 [73.2; 100.0]	90.0 [73.2; 100.0]	
RDT: No	53.4 [40.1; 71.0]	53.4 [40.1; 71.0]	0.62	81.7 [71.3; 93.5]	81.7 [71.3; 93.5]	0.96
Yes	65.2 [49.1; 86.4]	65.2 [49.1; 86.4]		81.0 [65.4; 100.0]	74.7 [57.4; 97.4]	
MRD: No	67.2 [56.5; 80.1]	67.2 [56.5; 80.1]	0.0011	85.4 [76.9; 94.9]	82.0 [71.9; 93.6]	0.23
Yes	16.7 [4.7; 59.1]	16.7 [4.7; 59.1]		65.6 [43.2; 99.7]	65.6 [43.2; 99.7]	
Age at diagnosis (y): <2.6	77.6 [63.3; 95.1]	77.6 [63.3; 95.1]	0.044	88.8 [77.6; 100.0]	88.8 [77.6; 100.0]	0.27
≥2.6	47.1 [34.2; 65.0]	47.1 [34.2; 65.0]		77.0 [65.1; 91.1]	73.3 [60.4; 89.0]	
Age at initiation (y): < 5	62.3 [50.3; 77.1]	62.3 [50.3; 77.1]	0.14	87.2 [78.7; 96.7]	83.2 [72.5; 95.5]	0.086
≥5	43.4 [24.9; 75.6]	43.4 [24.9; 75.6]		63.9 [43.9; 93.1]	63.9 [43.9; 93.1]	

**Table 3 cancers-15-02535-t003:** Univariate survival analysis using Cox models: Hazard ratios (HR), 95% confidence intervals, and *p*-values.

	EFS		OS	
	HR [95% CI]	*p*-Value	HR [95% CI]	*p*-Value
MYCN: A	Ref.	Ref.	Ref.	Ref.
NA	1.49 [0.61; 3.63]	0.38	0.73 [0.23; 2.35]	0.60
Chemotherapy cycles: 5	Ref.	Ref.	Ref.	Ref.
>5	0.96 [0.37; 2.50]	0.93	0.64 [0.18; 2.29]	0.49
ASCT: No	Ref.	Ref.	Ref.	Ref.
Yes	0.16 [0.02; 1.16]	0.070	0.40 [0.05; 3.08]	0.38
RDT: No	Ref.	Ref.	Ref.	Ref.
Yes	0.82 [0.38; 1.78]	0.62	1.03 [0.34; 3.08]	0.96
MRD: No	Ref.	Ref.	Ref.	Ref.
Yes	3.29 [1.54; 7.00]	0.0020	1.99 [0.62; 6.37]	0.24
Age at diagnosis (y)	1.13 [0.96; 1.32]	0.13	1.08 [0.82; 1.41]	0.59
Age at initiation (y)	1.12 [0.95; 1.31]	0.18	1.05 [0.79; 1.40]	0.72

Legend: ASCT = autologous stem cell transplant; RDT = radiotherapy; MRD = minimal residual disease.

## Data Availability

Data sharing is not applicable to this article as no new data were created or analyzed in this study.

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
