# Peer review of "Naxitamab Combined with Granulocyte-Macrophage Colony-Stimulating Factor as Consolidation for High-Risk Neuroblastoma Patients in First Complete Remission under Compassionate Use—Updated Outcome Report"

_cancers, 2023, doi:10.3390/cancers15092535_

Round 1
Reviewer 1 Report
The authors report on 82 patients with high-risk neuroblastoma in CR1 who received the anti-GD2-anitbody naxitamab that currently has market authorization in the US for the treatment of relapsed/refractory neuroblastoma. The issue is of interest for paediatric oncologist. The paper is well written and structured. However, it requires revisions before publication:
Major concern:
1. This is a single arm, non-controlled series of patients. The HSJD as treating institution had a contract with Ymabs as the manufacturer of Naxitamab. What was the strategy to avoid bias in patient selection and reporting? How many additional patients applied for treatment but were rejected at an early stage of application?
2. Naxitamab has no market authorization in the EU yet. The systematic assessment of efficacy and side effects of non-licenced drugs such as Naxitamab requires a fully approved clinical trial protocol and independent monitoring. Please give the clinicaltrials.gov identifier and the link to the trial protocol. If not available, this report cannot be published.
Minor concerns/corrections:
3. The authors should shorten the introduction. I recommend to omit the sequence on the Cheung and Heller paper from 1991.
4. What was the rational for giving oral docosahexaenoic acid triglyceride (DHA-TG) that has no market authorization in the EU as well? The reference [28] gives no additional explanation.
5. What was the interval (median and range) between diagnosis and start of Naxitamab/GM-CSF? It appears that most of the patients had more than 5 cycles of induction chemotherapy. Prolonged induction therapies and/or interval prior to Naxitamab/GM-CSF might exclude risk patients with early relapse.
6. Please give a list of induction chemotherapy regimens patients underwent. In the discussion section, the authors reveal, “all patients with MRD positive before immunotherapy were managed in centers following reported regimens from the 1990s”. The question arises to what extent the results of the analysis are applicable to state-of-the art neuroblastoma treatments.
7. What exactly means: “Patients who maintained complete remission at the end of consolidation (immunotherapy and radiotherapy) did not receive any further treatment.” Vice versa: What additional treatment was scheduled for those who did not?
8. Please add EFS and OS calculated from time of diagnosis.
9. Did the development of HAHA have an impact on outcome of the patients?
10. Apparently, the authors doubt on the efficacy of ASCT in the treatment of high-risk neuroblastoma patients in the era of anti-GD2-immunotherapy. They should at least state clearly that an answer to this question requires a powered comparative randomized clinical trial. Moreover, the authors should give the number of patients who underwent first ASCT during second-line treatment after relapse which also might contribute to the gap between EFS and OS.
Author Response
Reviewer #1
The authors report on 82 patients with high-risk neuroblastoma in CR1 who received the anti-GD2-anitbody naxitamab that currently has market authorization in the US for the treatment of relapsed/refractory neuroblastoma. The issue is of interest for paediatric oncologist. The paper is well written and structured. However, it requires revisions before publication:
Major concern:
- This is a single arm, non-controlled series of patients. The HSJD as treating institution had a contract with Ymabs as the manufacturer of Naxitamab. What was the strategy to avoid bias in patient selection and reporting? How many additional patients applied for treatment but were rejected at an early stage of application?
The contracts between HSJD and Ymabs did not influence the management of patients. Patients were unselected and were initially screened at our institution. Only those who were found to be in first complete remission received treatment as described. There was no other exclusion criteria in this study. Patients had received induction therapy in their local institutions and that was not controlled for. The starting point of the study was the confirmation of the first CR status done in our institution.
- Naxitamab has no market authorization in the EU yet. The systematic assessment of efficacy and side effects of non-licenced drugs such as Naxitamab requires a fully approved clinical trial protocol and independent monitoring. Please give the clinicaltrials.gov identifier and the link to the trial protocol. If not available, this report cannot be published.
The naxitamab 201 trial is a global, single-arm, open-label phase 2 trial with EudraCT No: 2017-001829-40 and IND No: 132793. ClinicalTrials.gov Identifier: NCT03363373.
Minor concerns/corrections:
- The authors should shorten the introduction. I recommend to omit the sequence on the Cheung and Heller paper from 1991.
The manuscript is short in number of words according to the editorial recommendation for research articles. There is no need to shorten any parts.
- What was the rational for giving oral docosahexaenoic acid triglyceride (DHA-TG) that has no market authorization in the EU as well? The reference [28] gives no additional explanation.
Some of the rationale:
The most highly concentrated Omega-3 PUFA in human tissues is by far DHA compared with ALA and EPA, being the membrane of the neurons the ones with a higher content of DHA, even surpassing the Omega-6 arachidonic acid (AA) content (Arterburn LM, 2006). DHA is the most abundant fatty acid in neural cells. It is incorporated in phospholipids of the cell membranes, thereby affecting membrane fluidity and protein activity (Stillwell W, 2003). Deficiency of DHA will lead to delayed neural development (Belkind-Gerson J, 2008). Compared to normal nervous tissue, neuroblastoma, glioma and meningioma, are profoundly deficient in DHA, whereas the level of the Omega-6 fatty acid arachidonic acid (AA) is increased (Reynolds LM, 2001; Martin DD, 1996; Kokoglu E, 1998). These suggest that an imbalance between Omega-3 and Omega-6 fatty acids may serve as an adaptation mechanism in nervous system tumors. DHA competes with AA for conversion by cyclooxygenases (COX) and lipooxygenases (LOX) to a wide range of lipid mediators, causing a plethora of physiological effects on the organism (Schmitz G, 2008). In fact, it is demonstrated that DHA induces apoptosis of neuroblastoma cells by mechanisms involving intracellular accumulation of cytotoxic DHA-derived metabolites (Lindskog M, 2006; Gleissman H, 2010) and treatment with DHA in murine xenografts with human neuroblastoma cells resulted in stable disease or partial response, depending on the DHA concentration (Gleissman H, 2011).
The required amount of Omega-3 intake is not clearly defined (U.S. Department of Agriculture ARS 2012) although there are certain recommendations for direct intake of 200 to 500 mg/day (FAO 2010; ISSFAL 2015; Nettleton JA, 2016) EPA + DHA for adult general health in the form of fish or fish oil, krill oil, or algae oil supplements based on individualized dietary patterns by state and age (Harris WS, 2009; Aranceta J, 2012). Doses generally exceeding 2 g/day combined EPA + DHA are needed to reduce prostaglandin E2 levels (Calder PC, 2013) and doses of 1 to 3.5 g/day combined EPA + DHA are most often used in the treatment of hypertriglyceridemia (Psota TL, 2006) or inflammatory disorders such as rheumatoid arthritis (Yates CM, 2014; Browning LM, 2012). A trial to test bioavailability into erythrocyte membranes of 5 groups of healthy volunteers (N=54) being supplemented for 30 days with increased daily doses of DHA-TG in each group, including one group with Placebo and the rest of groups with between 0,5g and 3,5g/day. Results in the membrane DHA concentration show a dose-dependent response (Carlos J. Contreras; Modificación del daño oxidativo en un grupo de ciclistas tras consumir ácido docosahexaenoico a distintas dosis; Tesis doctoral 2014; Universidad Católica de Murcia, Facultad de Ciencias de la Salud).
A high DHA intake should be obtained at a time of the day to facilitate a certain direct tumoricidal action and to ensure a sustained dietary treatment that limits the harmful effects of Omega-6. Also DHA intake should provide a preventive and favorable benefit for cell differentiation, apoptosis and synergy with chemotherapy and radiotherapy, as well as protection of non-tumor cells against these therapeutic agents (Hajjaji N, 2012). In many murine studies the doses of Omega-3 correspond to a diverse composition where DHA and EPA are part of the whole. This is why Gleissman and Hadajji are used as reference for this study, where the reference to DHA is clear and its concentration can vary between 0.5 and 1.5 g/kg/d. The reason is double, first to identify the specific effect of DHA and another because the Omega-3 formulation of the children's market offers very diverse compositions with different vehicles whose bioavailability is not always the most suitable, as expressed in a previous study of our group (Drobnic F, 2011). Thus, we will use a concentration adequate to the body weight and that the transport vehicle of the DHA is triglycerides and not an ethyl ester. The usual doses used in cardiovascular prevention and in inflammation in sports oscillate between 2 and 6 g/d for adult subjects, while there are no consistent adverse events in humans consuming up to 7.5 g/day of DHA (Lien EL, 2009). Taking into account all previously mentioned studies and that the weight of the patients diagnosed of neuroblastoma will oscillate between 20 and 40 kg, a dose of 0.25 g/kg is proposed. A fifth part of that used in a murine model (Gleissman H, 2011), being one of the doses 50% of the total and the other two 25%. It seems advisable also to maintain an administration every 8 h given that the half-life is somewhat over 6 hours (Gleissman H, 2011). Thus, a 20-40 kg child would take between 2 and 4 gr of DHA, of which half would be administered in a single oral intake and the rest in the other two administrations during the day, which can be matched with meals.
DHA is labeled as “food for special medical purposes” and not a medicinal product. The intervention is considered dietary supplementation. We included some of this rationale in the revised manuscript.
- What was the interval (median and range) between diagnosis and start of Naxitamab/GM-CSF? It appears that most of the patients had more than 5 cycles of induction chemotherapy. Prolonged induction therapies and/or interval prior to Naxitamab/GM-CSF might exclude risk patients with early relapse.
The number of cycles (<> 5) or duration of induction regimens was analyzed and found not be significant as shown in Table 2. As suggested by the reviewer, we added the time from diagnosis to treatment initiation, in months, in this cohort: median=8.7, range=(5.4, 21.5)
- Please give a list of induction chemotherapy regimens patients underwent. In the discussion section, the authors reveal, “all patients with MRD positive before immunotherapy were managed in centers following reported regimens from the 1990s”. The question arises to what extent the results of the analysis are applicable to state-of-the art neuroblastoma treatments.
The list of induction regimens used includes: mN7, GPOH NB2004, SIOPEN HR-NB01, COG 3973, China BCC protocols, GALOP protocol, Russian NB HR, and CCG. The Chinese and GALOP protocols were based upon 1990s regimens. These are current protocols used worldwide and reflects the reality of today’s management of children’s with HR-NB worldwide. The fact that MRD positivity was found mostly on patients having received older protocols provided the basis for the comparison supporting the hypothesis that intensified induction regiments provide better outcomes.
For western countries and more intensified regimens, MRD measurements are not used routinely therefore the relevance of MRD after induction remains unclear.
- What exactly means: “Patients who maintained complete remission at the end of consolidation (immunotherapy and radiotherapy) did not receive any further treatment.” Vice versa: What additional treatment was scheduled for those who did not?
Patients after completing planned treatment did not receive any further therapy (vaccine, DFMO or other strategies). Patients who relapse on treatment underwent a variety of rescue therapies.
- Please add EFS and OS calculated from time of diagnosis.
This request is not appropriate in the context of our study since some of the variables we considered (such as the number of induction chemotherapy cycles or ASCT) are not assigned at the time of diagnosis. Moreover, our intervention (naxitamab treatment) cannot be assigned at diagnosis either. Thus, performing survival analyses from the time of diagnosis would lead to the introduction of an immortal time bias, which is the insertion of a time-period in which the event of interest cannot occur. In reference to this statistical concept you can look up at:
. Immortal Time Bias in Observational Studies by Yadav K, et al. JAMA 2021;
.Immortal Time Bias in Observational Studies by Gleiss A, et al. Immortal Time Bias in Observational Studies.
In our study, it is clear that no events or deaths may occur between diagnosis and start of naxitamab because otherwise patients would not have entered the study.
If the reviewer is concerned about the time from diagnosis to treatment initiation, it has been added to the manuscript (see comment #5).
- Did the development of HAHA have an impact on outcome of the patients?
We could not analyze the impact of HAHA in this cohort given the low numbers. However, we recently reported that HAHA (ADA = anti-drug antibodies) did not impact on safety or efficacy of naxitamab in the larger cohort of the 201 trial (Kushner BH et al, ESMO Immuno Oncology meeting 2022).
- Apparently, the authors doubt on the efficacy of ASCT in the treatment of high-risk neuroblastoma patients in the era of anti-GD2-immunotherapy. They should at least state clearly that an answer to this question requires a powered comparative randomized clinical trial. Moreover, the authors should give the number of patients who underwent first ASCT during second-line treatment after relapse which also might contribute to the gap between EFS and OS.
The question of ASCT in the management of HR-NB has been in debate now for more than a decade and is not only from our group but others independently. Even now within the COG group there is an open debate on whether ASCT should be kept in the current regimen. The question (doubt) arises from the three randomized clinical trials already performed and published showing that ASCT does not improve OS of patients having received ASCT compared to does who did not. It does improve EFS but not OS. We published a thorough review on this subject (Mora J. Cancers 2022).
In the discussion we already state very clearly that more supportive data ins needed: “If these results could be reproduced in larger cohorts and more institutions”. At relapse none of the patients underwent ASCT since this is not recommended in our group.

Reviewer 2 Report
This is a retrospective study for high-risk neuroblastoma treated with anti-GD2 antibody and GM-CSF. The survival rate of the patients was excellent, but I have some questions.
1. Although this is a retrospective study, the detail of induction chemotherapy of patients are needed.
2. Please describe the treatment after first relapse of these patients, because there was a divergence between EFS and OS.
3. What is stage 4N? I recommend to use the INRG or INSS classification.
4. I think this study has many selection biases, so it is difficult to discuss the necessity of ASCT.
Author Response
Reviewer #2
This is a retrospective study for high-risk neuroblastoma treated with anti-GD2 antibody and GM-CSF. The survival rate of the patients was excellent, but I have some questions.
- Although this is a retrospective study, the detail of induction chemotherapy of patients are needed.
The list of induction regimens has been added in the revised manuscript as suggested by the reviewer.
- Please describe the treatment after first relapse of these patients, because there was a divergence between EFS and OS.
A variety of rescue regimens were used for relapsed patients depending on where the relapse occurred. For CNS only relapses the management follow the cRIT regimen. For systemic relapses patients received chemo immunotherapy following the HITS protocol. None of the patients in relapse received ASCT as rescue.
- What is stage 4N? I recommend to use the INRG or INSS classification.
4N is known as stage 4 Nodal because of limited lymph node dissemination. As the reviewer suggested we eliminated such categorization and followed INSS classification.
- I think this study has many selection biases, so it is difficult to discuss the necessity of ASCT.
Patients were unselected and were all screened at our institution upon arrival. Only those who were found and confirmed to be in first complete remission received treatment as described. There was no other exclusion criteria in this study. Patients had received induction therapy in their local institutions and that was not controlled for. From this starting point of the study there were no other biases that we could think of.
Round 2
Reviewer 1 Report
Thank you very much for the quick reply. I very much appreciate that most of my concern have been met perfectly. I regret that I have few remaining questions to the authors:
1. Thank you.
2. Thank you for clarification. However, trial NCT03363373 was entitled “Naxitamab for High-Risk Neuroblastoma Patients With Primary Refractory Disease or Incomplete Response to Salvage Treatment in Bone and/or Bone Marrow”. According to clinicaltrials.gov the trial was approved for neuroblastoma patients who did not achieve CR. Patients with CR were legally not eligible for the trial, at least according to the trial version made public at clincaltrials.gov. Perhaps the authors can clarify. Moreover, the trial number/link to the trial protocol must be found in the manuscript.
3. Accepted.
4. Thank you. Compounds prescribed to all trial patients must be part of the trial protocol.
5. Thank you.
6. Thank you. Please add this each induction regimen with corresponding patient number in table 1.
7. Thank you.
8. I do not agree. (1) The reported study is not an observational trial as defined by Yadav. (2) Naxitamab was part of the first-line therapy when given to patients in CR. Since the final outcome after the complete first-line therapy is of utmost interest of patients, parents, and physician, the authors must give EFS and OS from first diagnosis.
9. I recommend to revise the phrase “If these results could be reproduced in larger cohorts and more institutions…” to “If these results could be reproduced in prospective randomized clinical trials,…”.
Author Response
See attached document.

Reviewer 2 Report
The authors have responded appropriately to my comments and deserve to be published in this journal.
Author Response
See attached document
Round 3
Reviewer 1 Report
Thanks to the authors for quick reply. My personal concerns have been met. To clarify the regulatory background also for readers: (1) Please include “compassionate use” in the manuscript title. (2) Please include the NCI trial reference number and the information about compassionate use in the methods section.
Author Response
Thanks for the suggestions.- We added in the title and in the M&M section the compassionate use nature of the treatment.